# Describing historical habitat use of a native fish—cisco (*Coregonus artedi*)—in Lake Michigan between 1930 and 1932

Yu-Chun Kao[1¤]*, David B. Bunnell[2], Randy L. Eshenroder[3], Devin N. Murray[4]

**1** Department of Fisheries and Wildlife, Center for Systems Integration and Sustainability, Michigan State University, East Lansing, Michigan, United States of America, **2** U.S. Geological Survey Great Lakes Science Center, Ann Arbor, Michigan, United States of America, **3** Great Lakes Fishery Commission, Ann Arbor, Michigan, United States of America, **4** School for Environment and Sustainability, University of Michigan, Ann Arbor, Michigan, United States of America

¤ Current address: Eureka Aquatic Research LLC, Ann Arbor, Michigan, United States of America
* kyuchun@umich.edu

**Data Availability Statement:** All relevant data are within the manuscript and its Supporting Information files.

## Abstract

With the global-scale loss of biodiversity, current restoration programs have been often required as part of conservation plans for species richness and ecosystem integrity. The restoration of pelagic-oriented cisco (*Coregonus artedi*) has been an interest of Lake Michigan managers because it may increase the diversity and resilience of the fish assemblages and conserve the integrity of the ecosystems in a changing environment. To inform restoration, we described historical habitat use of cisco by analyzing a unique fishery-independent dataset collected in 1930–1932 by the U.S. Bureau of Fisheries' first research vessel *Fulmar* and a commercial catch dataset reported by the State of Michigan in the same period, both based on gear fished on the bottom. Our results confirmed that the two major embayments, Green Bay and Grand Traverse Bay, were important habitats for cisco and suggest that the Bays were capable of supporting cisco to complete its entire life cycle in the early 20th century as there was no lack of summer feeding and fall spawning habitats. Seasonally, our results showed that cisco stayed in nearshore waters in spring, migrated to offshore waters in summer, and then migrated back to nearshore waters in fall. The results also suggest that in summer, most ciscoes were in waters with bottom depths of 20–70 m, but the highest cisco density occurred in waters with a bottom depth around 40 m. We highlight the importance of embayment habitats to cisco restoration and the seasonal migration pattern of cisco identified in this study, which suggests that a restored cisco population can diversify the food web by occupying different habitats from the exotic fishes that now dominate the pelagic waters of Lake Michigan.

## Introduction

With the global-scale loss of biodiversity in the Anthropocene [1], restoration programs are commonly a required component of conservation planning that seeks to increase species

**Funding:** YCK, DBB, and DNM received funding for this study from U.S. Environmental Protection Agency's Great Lakes Restoration Initiative (https://www.glri.us/). The funder had no role in study design, data collection and analysis, decision to publish, or preparation of the manuscript.

**Competing interests:** The authors have declared that no competing interests exist.

richness and ecosystem integrity [2, 3]. For freshwater ecosystems, the restoration of native fishes could be a core component because fishes are not only functionally important in the food web but also economically important by supporting provisioning services [4]. To inform the restoration of native fishes, most frameworks delineate populations or stocks based on genetics and/or ecology [5, 6] and then assess existing biotic and/or abiotic threats [7–10], including the potential for habitat limitation [11–13].

The linkage between habitat and native fish restoration is based on the conservation principle that sustainable fish populations require a network of diverse habitats that support each life history stage [14], and, ideally, a diverse population structure across the landscape [15, 16]. Most of this research has been conducted in lotic ecosystems, with a focus on effects of increasing habitat connectivity [e.g., 17–19]. Relatively less research [e.g., 20, 21] has been conducted linking habitat to native fish restoration in lentic ecosystems, despite their vulnerability to anthropogenic alterations (e.g., draining of coastal wetlands, hardening of shorelines, nutrient or contaminant loading). As lake ecosystems naturally include a diversity of habitats including tributaries, coastal wetlands or embayments, nearshore littoral zones, and offshore profundal and pelagic zones, the underlying requirement for diverse and connected habitats to sustain fish populations parallels that of lotic ecosystems.

Managers across the Laurentian Great Lakes have prioritized restoration of native coregonines so as to increase the diversity and resilience of the fish assemblages and conserve the integrity of the ecosystems in a changing environment [22]. Coregonines were once abundant and diverse across the Great Lakes [23]. In Lake Michigan (Fig 1), for example, lake whitefish (*Coregonus clupeaformis*) and an assemblage of ciscoes (*Coregonus* spp.; subgenus *Leucichthys*) served as forage for native predators, such as lake trout (*Salvelinus namaycush*) and burbot (*Lota lota*) and supported commercial fisheries beginning in the 19th century [24].

As described by Koelz [26], the Lake Michigan assemblage of ciscoes comprised seven species of deepwater ciscoes (*C. alpenae, C. hoyi, C. johannae, C. kiyi, C. nigripinnis, C. reighardi,* and *C. zenithicus*) and one species of cisco (*C. artedi*) that occurred in shallower waters and shoals. Of the latter, Koelz [26] described cisco as the most widely distributed and diverse of the species in the basin—describing three subspecies—but only the "typical" *artedi* occurred in Lake Michigan. By the late 1960s, however, only one species of deepwater cisco (*C. hoyi*) persisted in Lake Michigan, and shallow-water cisco was extremely rare [24]. To avoid confusion, hereafter, we used the common name "cisco" (or ciscoes) for *C. artedi* including its subspecies and "deepwater ciscoes" for the assemblage of seven deepwater *Coregonus* species described by Koelz [26].

Several studies [24, 27, 28] have attributed the decline of cisco across the Great Lakes to overfishing, habitat degradation, and adverse effects of three exotic species: sea lamprey (*Petromyzon marinus*), rainbow smelt (*Osmerus mordax*), and alewife (*Alosa pseudoharengus*). Today, these stressors have been reduced in Lake Michigan because of management efforts, such as improvement of water quality [29], which has reduced hypoxia and presumably increased egg survival [30], suppression of sea lamprey [31], and reduction of rainbow smelt and alewife populations through predator stocking [32].

To determine whether habitat could yet limit cisco recovery, knowledge of historical habitat use for multiple life stages remains critical. What we know regarding historical habitat use of cisco in Lake Michigan is primarily based on accounts from the fishery, through descriptions by Koelz [26] and through historical commercial fishery records. These sources have depicted a cisco population that was concentrated in Green Bay (Fig 1). During 1946–1958, a 13-year period between the end of World War II and the collapse of cisco commercial fisheries, 88–95% of the catch occurred in Green Bay [33, 34]. Additional historical descriptions of habitat derive from Michigan Department of Natural Resources reports, which reported spawning runs of cisco in some Lake Michigan tributaries [35, 36].

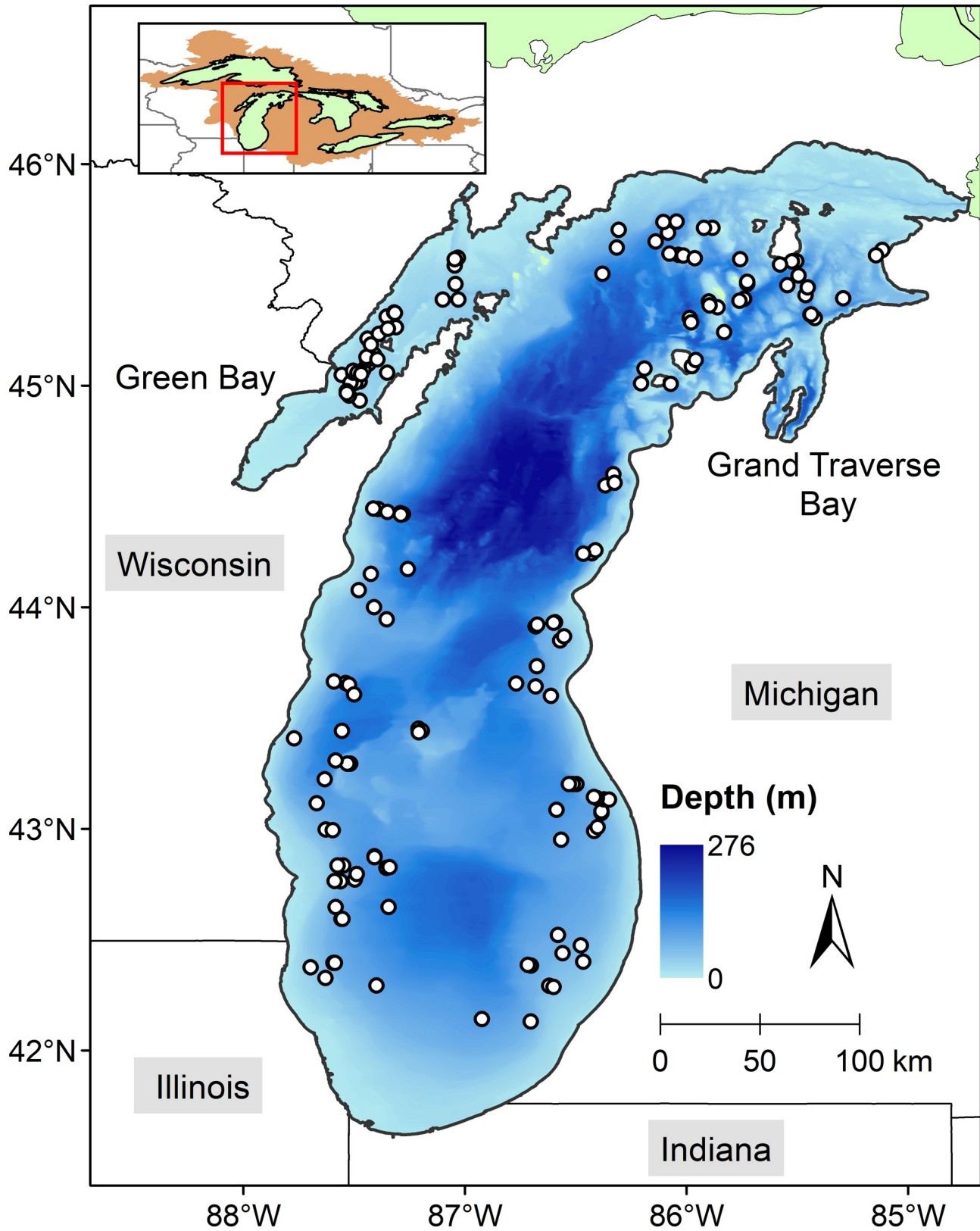

**Fig 1. Lake Michigan map.** The shaded area in the locator map depicts the Great Lakes watershed. White circles are the *Fulmar* sampling stations pooled across all seasons in 1930–1932. The map was generated by using a geographic information system (GIS) software ArcGIS version 10.4 (http://

www.esri.com/software/arcgis/). The GIS map layer for the *Fulmar* sampling stations was generated in this study. The GIS map layers for Lake Michigan bathymetry, political boundaries, the Great Lakes, and the Great Lakes watershed were obtained from Great Lakes Aquatic Habitat Framework's [25] public domain spatial database (https://glahf.org/data/).

To describe historical habitat use of cisco in Lake Michigan, we analyzed data from a unique fishery-independent dataset collected in 1930–1932 by the first vessel dedicated to aquatic research on the Great Lakes—the U.S. Bureau of Fisheries R/V *Fulmar* [37]—together with the commercial catch data reported to the State of Michigan in the same period. The *Fulmar* survey (Fig 2) was originally designed as an experiment to resolve a controversy among the four states around Lake Michigan over mesh-size limits for bottom-set gill nets employed in the fisheries of deepwater ciscoes. Although the original goal of establishing a common mesh-size limit was never achieved [38], the *Fulmar* investigation generated an extensive dataset, which has been analyzed to describe the growth, abundance, and community structure of various deepwater ciscoes [39–44] and to determine habitat use by lake trout [45], burbot [45], and deepwater ciscoes [46]. The State of Michigan started to collect and report monthly commercial catch and fishing effort by (fishery) statistical district in the Great Lakes in 1929 [47]. Spatially, these two datasets complement each other between 1930 and 1932, as the *Fulmar* survey did not cover Grand Traverse Bay while the commercial catch outside of State of Michigan waters were not reported by statistical district. Therefore, analyzing both of the *Fulmar* dataset and the spatially resolved commercial catch dataset could make it possible to describe the historical habitat use of cisco throughout Lake Michigan. The objectives in this study were to (1) describe cisco's historical habitat use by size class and season, (2) generate maps that predict historical distribution of cisco, and (3) document historical distribution of cisco when fish were expected to be gravid.

## Materials and methods

### Study area

Lake Michigan (Fig 1) is bordered by the states of Michigan, Wisconsin, Illinois, and Indiana and can be divided into three sub-basins: Green Bay (area 4,450 km$^2$, average depth 17 m), Grand Traverse Bay (area 720 km$^2$, average depth 55 m), and the main Basin (area 52,720 km$^2$, average depth 91 m). In terms of trophic status, southern Green Bay is still considered eutrophic, despite years of phosphorus abatement [48], while both Grand Traverse Bay and the main basin are considered oligotrophic [29, 49].

### Data

Data collected from the *Fulmar* surveys were recorded in notebooks (Fig 2C) and are now archived at the U.S. Geological Survey's Great Lakes Science Center. The *Fulmar* catch and biological data resulted from 363 bottom-set gill-net lifts distributed throughout the main basin and Green Bay between April and November in 1930–1932 (Fig 1). Each lift included 1–7 gangs of linen gill nets. Each gang comprised 3–5 panels each having a length of 155 m, a height of 1.5 m, and a (stretch-) mesh size of either 60, 64, 67, 70, or 76 mm. The information associated with sampling operations included date, location (latitude and longitude), bottom depth (starting and ending bottom depths of each gill-net set), and number of nights out. The digitization of the *Fulmar* data notebooks was started in the late 1990s [45] and finished in this study. We provide the *Fulmar* dataset in a supporting information file (S1 Datasets).

Summaries of monthly catch reports submitted by commercial fishermen to the State of Michigan were hand tabulated and archived as originals or microfiche at the U.S. Geological

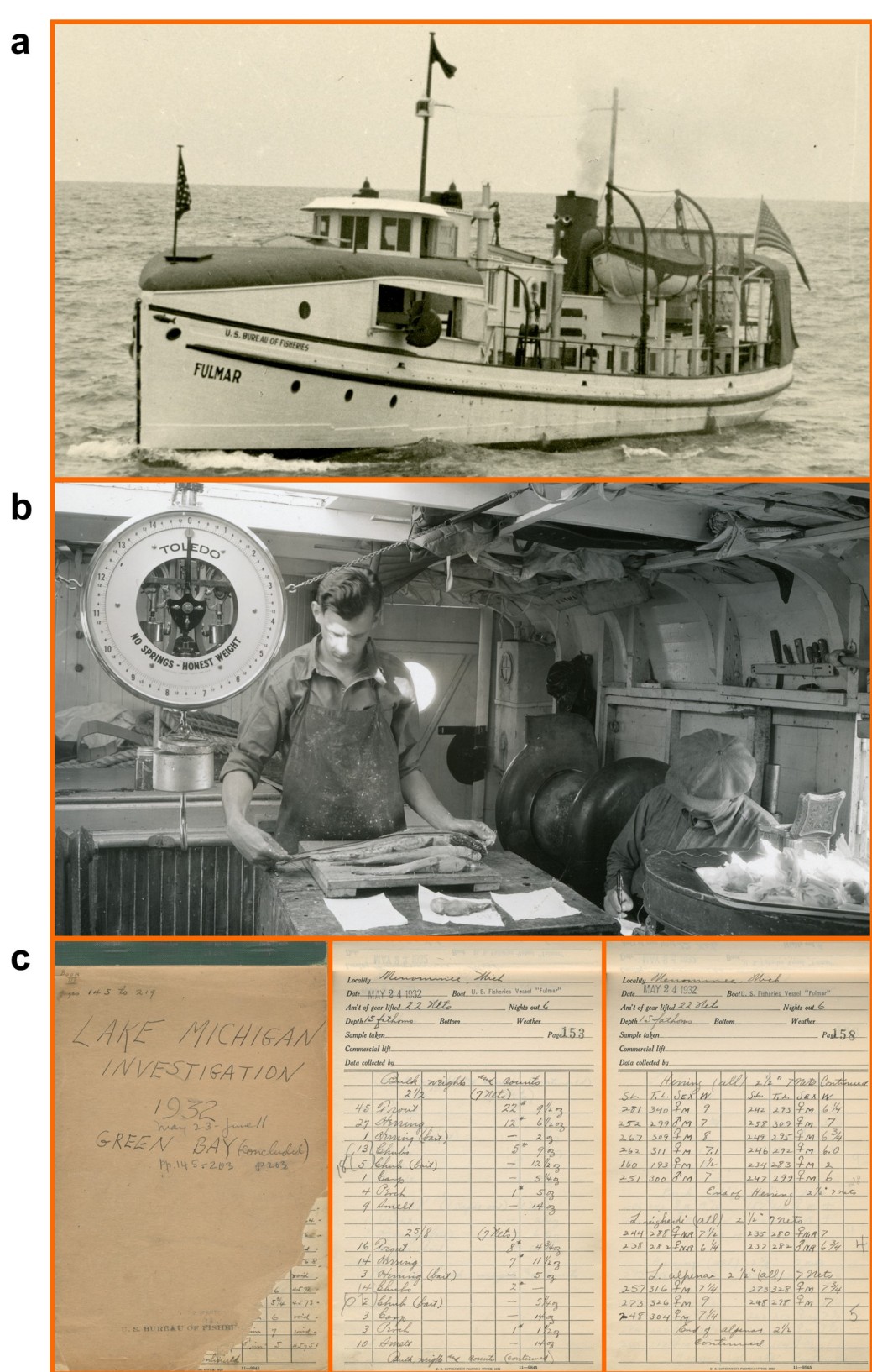

**Fig 2. Selected photos for the *Fulmar* survey (credit: U.S. Geological Survey).** (A) The U.S. Bureau of Fisheries research vessel *Fulmar*. (B) Researchers onboard taking and recording biological measurements. (C) Data notebooks. These

photos, which are archived at the U.S. Geological Survey's Great Lakes Science Center, are in public domain as they were taken by U.S. federal employees to show how U.S. federal employees carried out research.

Survey's Great Lakes Science Center. In the analyses, we used monthly catch and fishing effort for (fishery) statistical districts MM-1 to MM-8 (Fig 3). Although catch and effort were reported for eight gear types, we only used data for bottom-set gill nets (with mesh sizes of about 70-mm) and pound nets, which comprised the bulk (>93%) of annual cisco catch from the State of Michigan in 1930–1932, the period of study. We also provide the commercial catch dataset used in the analyses in a supporting information file (S1 Datasets).

## Statistical analyses for the *Fulmar* data

We used generalized additive models (GAMs) with a Tweedie error distribution and a natural logarithmic link function [50] to assess whether cisco catch in the *Fulmar* surveys varied with mesh size, season, and bottom depth, while adjusting for sampling effort and, if needed, spatial

**Fig 3. The four Lake Michigan regions defined in this study and the State of Michigan's (fishery) statistical districts (MM-1–MM-8).** The four regions, including Green Bay, Grand Traverse Bay, northern main basin, southern main basin, are marked in different colors and the State of Michigan's fishery statistical districts are cross-hatched. The map was generated by using a geographic information system (GIS) software ArcGIS version 10.4 http://www.esri.com/software/arcgis/). The GIS map layers for Lake Michigan and the State of Michigan's statistical districts were obtained from Great Lakes Aquatic Habitat Framework's [25] public domain spatial database (https://glahf.org/data/).

variability. We chose GAMs with a Tweedie error distribution because of its capability to model a count response variable (number of cisco caught in this study), even if it is over dispersed [50] as a non-linear function of categorical variables (e.g., mesh size, season), continuous variables (e.g., bottom depth), and spatial variables (latitude and longitude).

We then used Akaike's Information Criterion (AIC) to evaluate which of the 12 candidate GAMs used for modeling the *Fulmar* data (hereafter, *Fulmar* GAMs) were the most parsimonious (refer to Wood [50] for calculating AIC for GAMs). In the simplest *Fulmar* GAM, Model 1, cisco catch was a function of mesh size, season, and bottom depth, together with a function ($f_e$) for fishing effort adjusted in accordance with sampling duration (nights out) and the number of panels of gill nets with that mesh size

$$\text{Model 1}: \ln(Catch) = Mesh + Season + s_d(Depth) + f_e \tag{1}$$

where *Catch* is cisco catch in number of fish, *Mesh* is a categorical variable for mesh size, *Season* is a categorical variable for three seasons defined later, and $s_d(Depth)$ is a smoothing function for bottom depth (i.e., the average depth for the two ends of each gill-net set in m). The function $f_e$ was expressed as $f_e = s_n(Night) + o(LNP)$, where $s_n(Night)$ is a smoothing function for sampling duration in number of nights out and $o(LNP)$ is the natural logarithm of the number of gill-net panels included as an offset variable (i.e., a variable with a GAM coefficient constrained to one). We defined seasons as spring (April–June), summer (July–September), and fall (October–November) based on thermal stratification patterns observed in Lake Michigan during the *Fulmar* surveys [51]. As the pilot analyses showed clear differences in catch rates and size selectivity across gill nets with different mesh sizes, we included the categorical variable *Mesh* not only as an adjustment factor for the difference in catch rates across mesh sizes, but also as a proxy of cisco size class. Consequently, in terms of habitat use, Model 1 hypothesized that cisco had the same bottom-depth preference across size classes and seasons.

In the following three *Fulmar* GAMs, Models 2–4, we extended Model 1 by accounting for interactions between *Mesh* and *Depth*, between *Season* and *Depth*, or both

$$\text{Model 2}: \ln(Catch) = Mesh + Season + s_{md}(Mesh \times Depth) + f_e \tag{2}$$

$$\text{Model 3}: \ln(Catch) = Mesh + Season + s_{sd}(Season \times Depth) + f_e \tag{3}$$

$$\text{Model 4}: \ln(Catch) = Mesh + Season + s_{md} + s_{sd} + f_e \tag{4}$$

where $s_{md}(Mesh \times Depth)$ indicates that a smoothing function of bottom depth is fitted for each of the mesh sizes, $s_{sd}(Season \times Depth)$ indicates that a smoothing function of bottom depth is fitted for each of the three seasons, and, as used in Model 4, $s_{md}$ and $s_{sd}$ are shorthand notations for $s_{md}(Mesh \times Depth)$ and $s_{sd}(Season \times Depth)$, respectively. In terms of habitat use, Model 2 hypothesized that cisco's bottom-depth preference varied across size classes but not across seasons, Model 3 hypothesized that cisco's bottom-depth preference varied across seasons but not across size classes, and Model 4 hypothesized that cisco's bottom-depth preference varied across size classes and seasons.

In the next set of four *Fulmar* GAMs (Models 5–8), we extended Models 1–4 by adding a categorical variable *Basin* for the two sub-basins that were sampled—Green Bay and the main basin of Lake Michigan

$$\text{Model 5}: \ln(Catch) = Mesh + Season + Basin + s_d + f_e \tag{5}$$

$$\text{Model 6}: \ln(Catch) = Mesh + Season + Basin + s_{md} + f_e \tag{6}$$

$$\text{Model 7}: \ \ln(Catch) = Mesh + Season + Basin + s_{sd} + f_e \tag{7}$$

$$\text{Model 8}: \ \ln(Catch) = Mesh + Season + Basin + s_{md} + s_{sd} + f_e \tag{8}$$

where $s_d$, as used in Model 5, is the shorthand notation for $s_d(Depth)$. In addition to what Models 1–4 hypothesized for cisco's habitat preference, Models 5–8 further hypothesized that cisco preferred Green Bay over the main basin, or vice versa.

In the final set of four *Fulmar* GAMs (Models 9–12), we extended Models 5–8 to account for spatial variability by adding a Gaussian-process smoothing function $s_g(Lat, Lon)$

$$\text{Model 9}: \ \ln(Catch) = Mesh + Season + Basin + s_d + s_g(Lat, Lon) + f_e \tag{9}$$

$$\text{Model 10}: \ \ln(Catch) = Mesh + Season + Basin + s_{md} + s_g + f_e \tag{10}$$

$$\text{Model 11}: \ \ln(Catch) = Mesh + Season + Basin + s_{sd} + s_g + f_e \tag{11}$$

$$\text{Model 12}: \ \ln(Catch) = Mesh + Season + Basin + s_{md} + s_{sd} + s_g + f_e \tag{12}$$

where *Lat* is latitude, *Lon* is longitude, and, in Models 10–12, $s_g$ is the shorthand notation for $s_g(Lat, Lon)$. In addition to what Models 5–8 hypothesized for cisco's habitat preference, Models 9–12 further hypothesized that cisco preferred certain regions within each sub-basin (i.e., within Lake Michigan main basin and/or within Green Bay).

We used the package "mgcv" version 1.8–28 [52] in R version 3.6.1 [53] to perform the GAM analyses. In the package, smoothing functions are estimated using nonparametric, penalized-spline methods, in which the degrees of freedom of each smoothing function are estimated by fitting to observations. We used the thin-plate regression spline [50] as the smoothing basis for functions $s_d$, $s_{md}$, $s_{sd}$, and $s_n$. We set the Gaussian-process smoothing function $s_g$ to have a spherical correlation structure [50], in which spatial auto-correlation was precluded if two points were separated by >90 km, a value based on the maximum migration distance for cisco observed in Lake Michigan between 1929 and 1931 [54]. We used the restricted-maximum-likelihood method to estimate parameters and smoothing functions in all 12 *Fulmar* GAMs [50]. For every *Fulmar* GAM, we calculated ΔAIC as the difference between its AIC and the minimum AIC across all 12 *Fulmar* GAMs. We reported estimated degrees of freedom, ΔAIC, and the deviance explained each *Fulmar* GAM and considered the GAMs with ΔAIC <4 as the most parsimonious [46, 55].

## Statistical analyses for the commercial catch data

Because the commercial catch dataset did not include as much spatiotemporal information associated with fishing operations as the *Fulmar* dataset, we developed only one commercial-catch GAM (Model 13) that incorporated all available information that could be used to derive GAM predictor variables. Specifically, our commercial-catch GAM was based on a gamma error distribution and a natural logarithmic link function [50] to assess whether the commercial cisco catch varied with season and region, while adjusting for fishing gear and effort. This commercial-catch GAM (Model 13) was expressed as

$$\text{Model 13}: \ \ln(ComCatch) = Season + Region + Gear + s_e(Gear \times Effort) \tag{13}$$

where *ComCatch* is commercial cisco catch in tonnes, *Season* is a categorical variable as

defined in Eq (1), *Region* is a categorical variable for the four regions (Fig 3), *Gear* is a categorical variable for gill nets and pound nets, *Effort* is fishing effort reported in total gill-net length (km) or number of pound-net lifts, and $s_e(Gear \times Effort)$ indicates that a smoothing function of fishing effort is fitted for each gear type. We chose a gamma error distribution because the response variable (*ComCatch*) was continuous, positive, and right skewed. We divided the main basin into two regions at 44˚N and included a total of four regions in the analyses: Green Bay, Grand Traverse Bay, northern main basin, and southern main basin (Fig 3). Among the State of Michigan's statistical district, MM-1 data were used for Green Bay, MM-4 data were used for Grand Traverse Bay, lumped MM-3 and MM-5 data were used for the northern main basin, and lumped MM-7 and MM-8 data were used for the southern main basin (Fig 3). Due to very low fishing efforts, which were zero in most months during the study period, MM-2 and MM-6 data were not included in the analyses. As before, we used the package "mgcv" version 1.8–28 [52] in R version 3.6.1 [53] to perform the GAM analyses, in which we used the thin-plate regression spline as the smoothing basis for the function $s_e$ and the restricted-maximum-likelihood method to estimate parameters and $s_e$ [50].

## Maps for historical cisco distribution

We used the geographic information system (GIS) software ArcGIS version 10.4 ([http://www.esri.com/software/arcgis/](http://www.esri.com/software/arcgis/)) to generate the maps for cisco distribution in Lake Michigan in 1930–1932. The cisco distribution was indicated by the predictions of the selected *Fulmar*-GAM for cisco catch. To generate the GIS map layers for predicted cisco catch, we obtained the *Fulmar*-GAM inputs for Lake Michigan bathymetry from Great Lakes Aquatic Habitat Framework's [25] public domain spatial database ([https://glahf.org/data/](https://glahf.org/data/)), which included information for latitude, longitude, and bottom depth at a grid resolution of $30 \times 30$ m$^2$. Due to no *Fulmar* survey being conducted in Grand Traverse Bay (GTB), the predicted cisco catch in this sub-basin was based on the results of the statistical analyses, which could be one of the following three scenarios:

1. The selected *Fulmar* GAM is one of Models 1–4: This allows us to predict GTB catch without making any assumption.

2. The selected *Fulmar* GAM is one of Models 5–8: This allows us to predict GTB catch by treating GTB as either Green Bay or the main basin, which could be informed by results of the commercial-catch GAM (Model 13) that was fitted to data collected in Green Bay, GTB, and the main basin.

3. The selected *Fulmar* GAM is one of Models 9–12: This allows us to predict GTB catch by using the corresponding model in Models 5–8 and following the same method as described in scenario 2. Specifically, Models 5–8 are the same as Models 9–12 without the Gaussian-process smoothing function $s_g$. A model including $s_g$ to extrapolate catch in GTB could not be used because it is not reasonable to assume that cisco preferred the same region in GTB as in either Green Bay or the main basin. For example, it is not reasonable to assume that cisco preferred northern GTB because they preferred northern Green Bay.

## Historical distribution of cisco in the gravid season

We used ArcGIS version 10.4 to generate a map for historical distribution of cisco in the gravid season. Due to data limitations, we could not achieve our initial target of generating a map for historical distribution of cisco in the spawning season. The *Fulmar* survey categorized maturity into one of the five stages: immature, mature, gravid (or nearly ripe), ripe, and spent. However, no cisco was recorded as ripe or spent, which precluded the generation a map of historical

distribution of cisco in the spawning season. In the map for historical distribution of cisco in the gravid season, we generated a GIS map layer to show the *Fulmar* stations with cisco presence or absence during the gravid season. The gravid season was defined as beginning on the earliest date a gravid cisco was caught and ended at the latest date (November 19) for a *Fulmar* lift.

## Results

### *Fulmar* data summary

The *Fulmar* survey consisted of 361 gill-net lifts made between June 26 and November 15 in 1930, between May 6 and November 19 in 1931, and between April 21 and September 13 in 1932. Although the combined catch of cisco and deepwater ciscoes was recorded from every lift, the catch was not always enumerated by species. As a result, catch data for cisco were available only for 291 of the 361 lifts. Among the 291 lifts that could be included in the statistical analyses, we excluded data from 76-mm mesh in 176 lifts due to low catch rates—only three ciscoes were caught in panels of this mesh size. We also excluded data from three summer lifts in 1930 with sampling durations of 16–18 nights. These gangs were retrieved late due to engine malfunction as described in the *Fulmar's* scientific logs, which were also archived at U.S. Geological Survey's Great Lakes Science Center.

Finally, we included data from gangs in 288 lifts in the statistical analyses (Table 1). These lifts were made in Green Bay and the main basin in all three seasons, although the number of lifts made in Green Bay in fall was relatively small (N = 4). These lifts were made at comparable bottom depths across seasons, ranging from 24 to 142 m in spring, from 18 to 155 m in summer, and from 29 to 170 m in fall. A total of 10,867 ciscoes were caught in these lifts, total lengths were measured for 4,007 ciscoes, and maturity was recorded for 3,304 ciscoes. The size of caught ciscoes increased with the mesh size. The average total length ± standard deviation of ciscoes from 60-mm mesh was 280 ± 17 mm (N = 1,299), from 64-mm mesh was 300 ± 23 mm (N = 1,495), from 67-mm mesh was 306 ± 29 mm (N = 904), and from 70-mm mesh was 325 ± 34 mm (N = 309).

### Statistical analyses for the *Fulmar* data

Models 11 and 12 were the most parsimonious (i.e., $\Delta AIC < 4$) among all 12 candidate GAMs, but Model 11 was judged to be superior because its estimated degrees of freedom were smaller than those of Model 12 (Table 2). Model 11 explained 78% of the deviance and every term in the model was significant at $p < 0.001$ level, except for the smoothing functions of bottom depth for spring (p = 0.672) and fall (p = 0.028) and the Gaussian-process smoothing function for Green Bay (p = 0.003). With adjustments for sampling effort and spatial variability, Model 11 showed that cisco catch was decreasing with increasing mesh size (Fig 4A), was higher in summer and fall than in spring (Fig 4B), was higher in Green Bay than in the main basin (Fig 4C), was decreasing with bottom depth in spring and fall (Fig 4D and 4F), and was increasing with bottom depth when it was less than about 40 m but decreasing with bottom depth when it was greater than about 40 m (Fig 4E). These results support the following hypotheses for

**Table 1. Number of bottom-set gill-net lifts included in the statistical analyses.**

| Season | Sub-basin | | Total |
|---|---|---|---|
| | **Green Bay** | **Main basin** | |
| Spring | 27 | 55 | 82 |
| Summer | 11 | 152 | 163 |
| Fall | 4 | 39 | 43 |

**Table 2. Results for the selection of Generalized Additive Models (GAM) fitted to the *Fulmar* data.**

| | | | | | | | | | ΔAIC | edf | Deviance explained (%) |
|---|---|---|---|---|---|---|---|---|---|---|---|
| **Basic models** | | | | | | | | | | | |
| Model 1: | ln(*Catch*)~ | *Mesh* | + *Season* | | + $s_d$ | | | + $f_e$ | 101.1 | 16.9 | 72.5 |
| Model 2: | ln(*Catch*)~ | *Mesh* | + *Season* | | + $s_{md}$ | | | + $f_e$ | 121.0 | 27.1 | 72.4 |
| Model 3: | ln(*Catch*)~ | *Mesh* | + *Season* | | | + $s_{sd}$ | | + $f_e$ | 86.5 | 24.9 | 74.0 |
| Model 4: | ln(*Catch*)~ | *Mesh* | + *Season* | | + $s_{md}$ | + $s_{sd}$ | | + $f_e$ | 84.9 | 29.3 | 74.4 |
| **Models accounting for sub-basin preference** | | | | | | | | | | | |
| Model 5: | ln(*Catch*)~ | *Mesh* | + *Season* | + *Basin* | + $s_d$ | | | + $f_e$ | 79.3 | 16.6 | 73.5 |
| Model 6: | ln(*Catch*)~ | *Mesh* | + *Season* | + *Basin* | + $s_{md}$ | | | + $f_e$ | 81.3 | 19.0 | 73.6 |
| Model 7: | ln(*Catch*)~ | *Mesh* | + *Season* | + *Basin* | | + $s_{sd}$ | | + $f_e$ | 35.6 | 19.9 | 75.7 |
| Model 8: | ln(*Catch*)~ | *Mesh* | + *Season* | + *Basin* | + $s_{md}$ | + $s_{sd}$ | | + $f_e$ | 33.7 | 24.4 | 76.2 |
| **Models accounting for sub-basin preference and spatial variability** | | | | | | | | | | | |
| Model 9: | ln(*Catch*)~ | *Mesh* | + *Season* | + *Basin* | + $s_d$ | | + $s_g$ | + $f_e$ | 41.1 | 27.5 | 76.1 |
| Model 10: | ln(*Catch*)~ | *Mesh* | + *Season* | + *Basin* | + $s_{md}$ | | + $s_g$ | + $f_e$ | 44.0 | 29.9 | 76.1 |
| Model 11: | ln(*Catch*)~ | *Mesh* | + *Season* | + *Basin* | | + $s_{sd}$ | + $s_g$ | + $f_e$ | 2.6 | 30.3 | 78.0 |
| Model 12: | ln(*Catch*)~ | *Mesh* | + *Season* | + *Basin* | + $s_{md}$ | + $s_{sd}$ | + $s_g$ | + $f_e$ | 0.0 | 35.7 | 78.5 |

ΔAIC, difference between the Akaike's information criterion (AIC) of a GAM and the AIC of Model 12, which had the minimum AIC across the 12 candidate GAMs; edf, estimated degrees of freedom, the total degrees of freedom of all categorical variables and smoothing functions; *Catch*, catch in number of cisco; *Mesh*, categorical variable of mesh size (60, 64, 67, or 70 mm); *Season*, categorical variable of season (spring, summer, or fall); *Basin*, categorical variable of sub-basin (Green Bay or the main basin); $s_d$, smoothing function of bottom depth; $s_{md}$, smoothing function of bottom depth by mesh size; $s_{sd}$, smoothing function of bottom depth by season; $s_g$, Gaussian-process smoothing function, which accounts for spatial auto-correlation; $f_e$, fishing-effort adjustment function of number of nights out and number of gill-net panels.

historical habitat use by cisco in Lake Michigan: (1) cisco preferred Green Bay over the main basin and (2) the bottom-depth preference of cisco varied across seasons but not across size classes. Model 11 also included significant Gaussian-process smoothing functions (p = 0.003 for the Green Bay function and p < 0.001 for the main basin function), which supports the hypothesis that cisco preferred certain regions within Green Bay and within the main basin.

## Statistical analyses for the commercial catch data

The commercial-catch GAM had 12.2 estimated degrees of freedom and explained 87% of the deviance, and every term in the model was significant at p < 0.001 level. With adjustments for fishing effort, the model showed that commercial cisco catch was higher in spring and fall than in summer (Fig 5A), higher in Green Bay and Grand Traverse Bay than in the main basin, and higher in the southern than in the northern main basin (Fig 5B).

## Maps for historical cisco distribution

We generated maps for historical cisco distribution in Lake Michigan by season (Fig 6) by using predicted cisco catch based on the most parsimonious *Fulmar* GAM. As the commercial-catch GAM suggests that cisco catch in Grand Traverse Bay was more similar to the catch in Green Bay than in the main basin, cisco catch in Grand Traverse Bay was predicted by using Model 7 and treating Grand Traverse Bay as Green Bay.

## Historical distribution of cisco in the gravid season

A total of six gravid ciscoes were recorded in the *Fulmar* data and were caught on October 16 (N = 1), October 30 (N = 1), November 5 (N = 3), and November 6 (N = 1). Hence, October

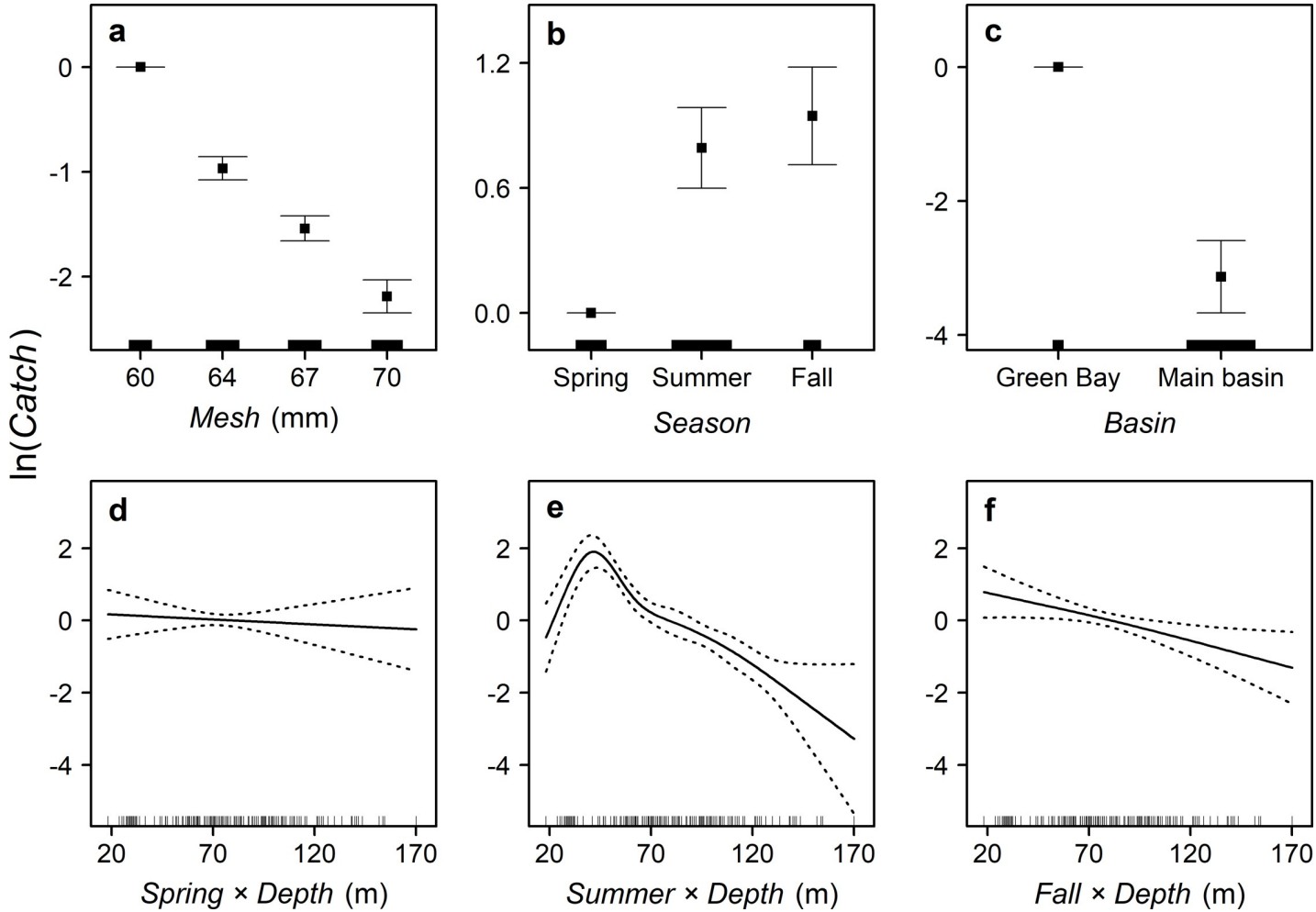

**Fig 4. The relationships between number of cisco caught (*Catch*) and gill-net mesh size (*Mesh*), season (*Season*), habitat type (*Basin*), and bottom depth (*Depth*), with adjustments for sampling effort and spatial variability, based on the selected generalized additive model fitted to the *Fulmar* data.** The points in panels a–c and solid lines in panels d–f represent the predicted values. The error bars in panels a–c and dotted lines in panels d–f represent +/− one standard error. Ticks above the horizontal axis represent the relative sample size in panels a–c and the distribution of data points in panels d–f.

16 was defined as the beginning of the gravid season. From October 16 to November 19, the latest date for a *Fulmar* lift, a total of 124 ciscoes were caught in 13 of 30 gill-net lifts. However, maturity was recorded for only 15 (6 gravid and 9 mature) of the 124 ciscoes caught in just 4 of the 13 gill-net lifts producing cisco. As shown in Fig 7, the 13 gill-net lifts with cisco were located in the central-western and southwestern main basin. No gill-net lifts were made in Green Bay and in a large area of northern main basin from October 16 to November 19 during the *Fulmar* survey.

## Discussion

### Historical habitat use of cisco

The fishery-independent data from the *Fulmar* supported previous findings from Smith [34] and Hile and Buettner [47], based on commercial catch records, that cisco was markedly more abundant in Green Bay than in the main basin of Lake Michigan in all seasons. One somewhat surprising finding, however, was that when accounting for effort, commercial catch in Grand

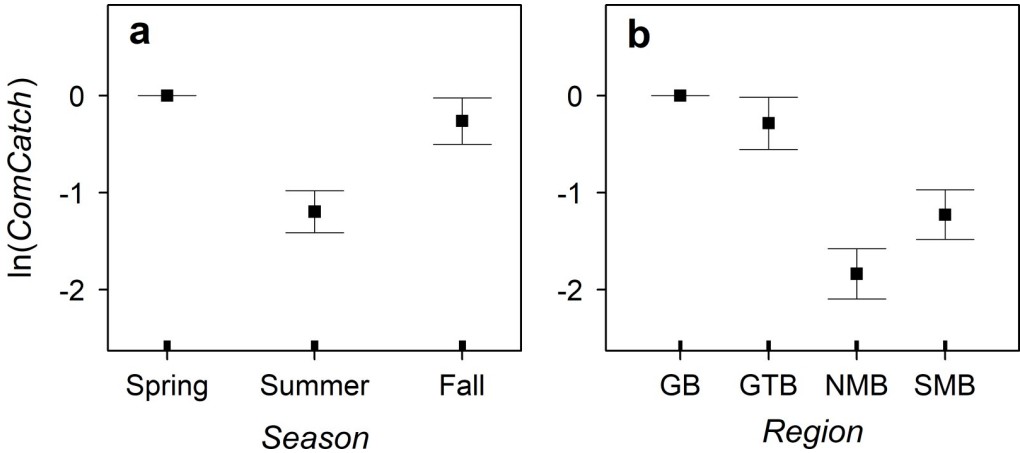

**Fig 5. The relationships between commercial cisco catch in tonnes (*ComCatch*) and season (*Season*) and Lake Michigan region (*Region*), with adjustments for gear type and fishing effort, based on the generalized additive model fitted to the State of Michigan's commercial catch data collected in 1930–1932.** The points represent the predicted values and the error bars represent +/− one standard error. Ticks above the horizontal axis represent the relative sample size.

Traverse Bay was just as high as in Green Bay historically, which reinforces the importance of embayments to cisco populations.

In addition, the *Fulmar* data indicated that in Lake Michigan, cisco migrated to deeper waters in summer, and inhabited shallower waters in spring and in fall. This pattern of shifting to deeper waters in summer is consistent with historical observations by Lake Michigan fishers

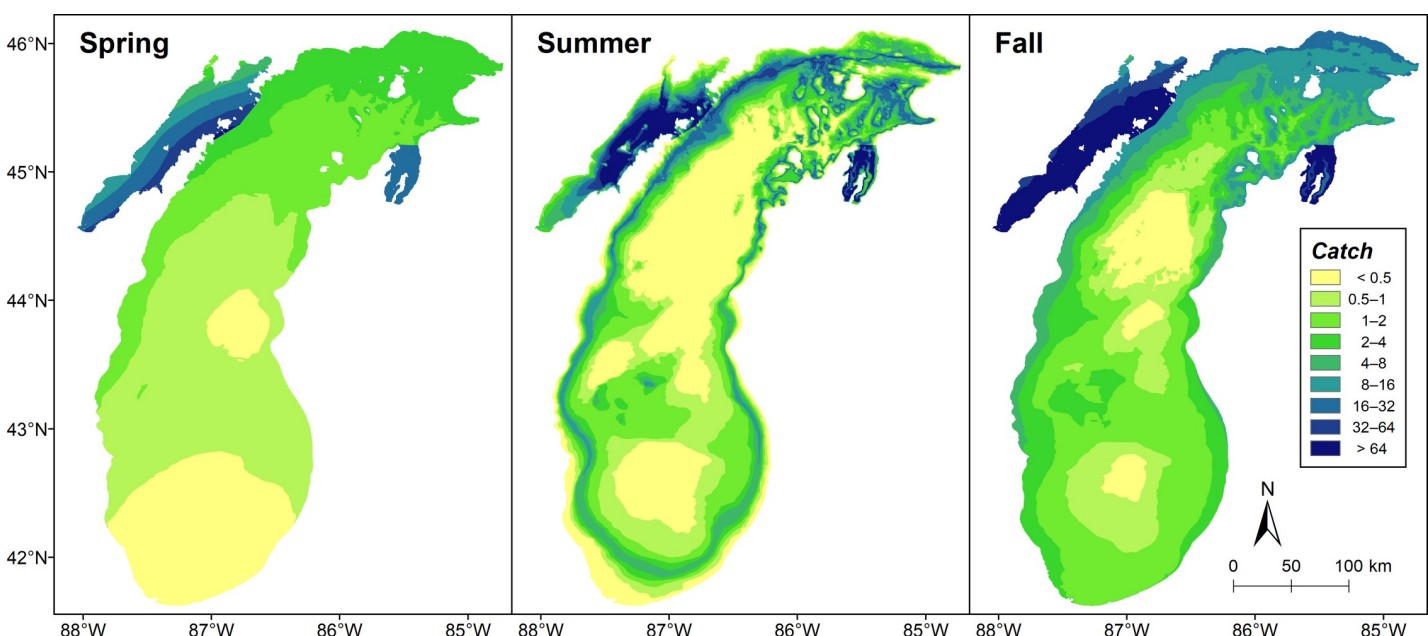

**Fig 6. Maps for historical cisco distribution in Lake Michigan by season, 1930–1932.** Cisco catch was predicted by using the selected generalized additive model fitted to the *Fulmar* data and was standardized to per panel of gill net with a 60-mm mesh size and a sampling duration of four nights. The maps were generated by using a geographic information system (GIS) software ArcGIS version 10.4 (http://www.esri.com/software/arcgis/). The GIS map layers for predicted cisco catch were generated in this study. The GIS map layers for Lake Michigan was obtained from Great Lakes Aquatic Habitat Framework's [25] public domain spatial database (https://glahf.org/data/).

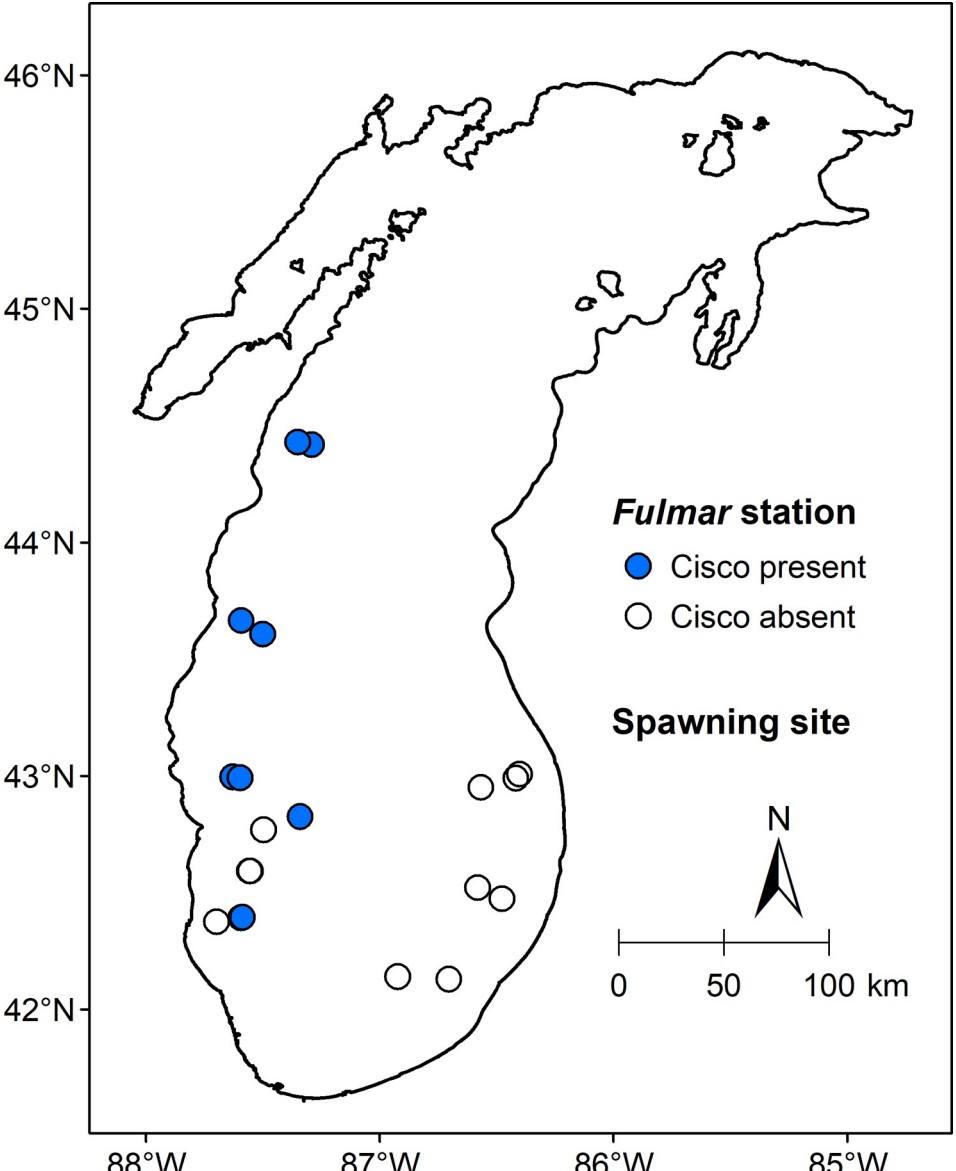

**Fig 7. Historical distribution of cisco in the gravid season in Lake Michigan.** Circles are *Fulmar* stations sampled from October 16, the earliest date when a gravid cisco was caught, to November 19, the latest date for a *Fulmar* lift between 1930 and 1932. Closed circles are sampling stations where ciscoes were caught, and open circles are sampling stations where no cisco was caught. The map was generated by using a geographic information system (GIS) software ArcGIS version 10.4 (http://www.esri.com/software/arcgis/). The GIS map layer for historical distribution of cisco in the gravid season was generated in this study. The GIS map layers for Lake Michigan was obtained from Great Lakes Aquatic Habitat Framework's [25] public domain spatial database (https://glahf.org/data/).

[26]. The results suggest, too, that cisco was mostly distributed in waters with bottom depths between 20 and 70 m, and the highest cisco density occurred in waters with bottom depth around 40 m in summer. Again, this finding is consistent with historical observations by fishers, who reported that cisco was caught in summer at bottom depths between 20–45 m across Lake Michigan [26]. One plausible explanation for this pattern of seasonal migrations is that cisco actively searched for water temperatures close to 10°C, the preferred thermal optimum [56], and avoided water temperatures greater than 17°C [34, 56, 57]. Given that the

thermocline in summer typically intersects with the bottom of the lake at between 20 and 40 m, cisco likely dispersed into these deeper waters to avoid warmer epilimnetic waters and to locate waters closer to 10˚C [51].

The results provided strong evidence that Green Bay and Grand Traverse Bay (hereafter, the Bays) were historically important habitats. Although local fishers had long speculated that cisco migrated from Green Bay to the main basin in spring [58], some areas in the Bays could have suitable habitats for cisco in summer. In Green Bay, about 15% of the surface area is between 30 and 50 m in depth (Fig 1), and Grand Traverse Bay offers abundant deeper waters relatively nearshore [see discussion in 59]. Early studies in Green Bay also showed that at least some cisco remained in Green Bay over summer. For example, Koelz [26] reported that cisco was abundant there in August 1920, and Smith [34] caught cisco in Green Bay with suspended gill nets throughout May to October 1952. Additionally, Goodyear et al. [60] documented that spawning habitats were available for cisco in both Bays. All together, these suggest that the Bays were capable of supporting cisco to complete its entire life cycle in the early 20th century.

The results also hint that habitat degradation had already occurred in Green Bay by the 1930s. In fall, the selected *Fulmar* GAM indicated that cisco preferred shallower waters but the extrapolated prediction showed that cisco was more abundant in the deeper outer Green Bay than in the shallower inner Green Bay. Pollution was reported as a factor affecting fish habitat in Lake Michigan as early as in the late nineteenth century [61]. In inner Green Bay, low dissolved oxygen (<1 mg/L) conditions under ice were first reported in 1938 and were associated with large inputs of nutrients and organic matters from the watershed where paper-making was concentrated [62]. However, the hypoxia problem could have lasted for years, but was not observed until 1938 when a systematic water-quality survey was first conducted in Green Bay [62]. Wells and McLain [24] also suggested that habitat degradation in Green Bay could have been a factor contributing to the marked decline of cisco commercial catch between around 1900 and 1920.

## Sampling biases

Bottom-set gill nets are not the ideal gear for sampling fish like cisco that often inhabit the pelagic waters of large lakes. Much of current understanding on the habitat of Great Lakes cisco populations derives from fishery-independent data in Lake Superior, where cisco remains relatively abundant in both embayments and offshore waters. Early studies by Dryer [63] and Selgeby [64] both reported low cisco catch rates in bottom-set gill nets except during the fall spawning season. Stockwell et al. [65] hypothesized that cisco in Lake Superior became more pelagic-oriented at larger sizes. This hypothesis was supported by Gorman et al. [66], who showed that both small (total length <185 mm) and large (total length >250 mm) ciscoes exhibited diel vertical migration, and during the day, large ciscoes inhabited the deep- and mid-hypolimnetic zones while small ciscoes were bottom-orientated. The diel vertical migration of cisco in Lake Superior was associated with the capture of *Mysis diluviana* for food and the avoidance of lake trout predation [67, 68]. As both *Mysis diluviana* and lake trout were historically abundant in Lake Michigan [24], large ciscoes (total length >250 mm; which comprised 98% of the *Fulmar* catch) in historical Lake Michigan could have similar diel vertical migration patterns as observed contemporarily in Lake Superior. Therefore, the finding of lower cisco catch rates with increasing mesh sizes could be explained by larger, less-abundant ciscoes being more pelagic and the predicted distribution in fall could be the least biased given the higher catch rates from bottom-set gill nets in the season.

The behavior and habitat preference of cisco might not be the same in Lake Michigan and Lake Superior. Koelz [26], however, reported that cisco in Lake Michigan in summer could be

pelagic as in Lake Superior, although cisco in Lake Superior could spend more time near the surface due to cooler temperatures. After adjusting for mesh size and the other variables, the selected *Fulmar* GAM showed that average cisco catch was only 16% lower in summer than in fall, suggesting that the large ciscoes in Lake Michigan still encountered bottom-set gill nets at comparable rates in summer and in fall. While the selected *Fulmar* GAM showed that average cisco catch was 63% lower in spring than in fall, the commercial-catch GAM showed that average catch in spring and fall were about the same. One possible reason for the low spring catch of cisco in the *Fulmar* data was that the nets were set too deep. In the 1930s, commercial pound nets were set at waters around 11 m in depth and commercial bottom-set gill nets were set at waters of all depths [34]. As about 74% of the commercial cisco catch was from pound-net fisheries in 1930–1932, it was assumed that, more likely, cisco inhabited waters <18 m in depth in the spring, where the *Fulmar* survey did not sample.

In the main basin of Lake Michigan, relative catch rates in the north and south revealed opposite patterns in the *Fulmar* (more abundant in the north) and commercial catch (more abundant in the south) data. This discrepancy may have been caused, in part, by the differences in fishing depths as the *Fulmar* survey typically fished in deeper waters (targeting deep-water ciscoes) than the commercial cisco fisheries. Interestingly, the commercial catch data available by county across all four Lake Michigan states in 1903 [69] suggest that cisco was abundant throughout the main basin. However, historical commercial catch reports from the State of Michigan (archived at the U.S. Geological Survey's Great Lakes Science Center) showed that cisco fisheries collapsed around 1940 in the southern main basin and then around 1960 in more northerly waters.

## Implications for spawning habitat

The *Fulmar* data contained limited information about cisco spawning habitat because the field season (mid-April to mid-November) did not extend into the height of the spawning season, usually between mid-November and early December in Lake Michigan [26], and the sampling stations in fall (all > 29 m in bottom depth) did not extend to shallower waters where cisco spawning was reported [59]. As cisco did not migrate extensively [54], the distribution of cisco in the gravid season (Fig 7) could confirm the reported spawning sites in central-western and southwestern waters of the main basin [60]. Although there was no cisco caught during the gravid season at the seven stations in the southeastern main basin, the small number of lifts (N = 7) may have limited the ability to confirm reported spawning sites in this region [60]. Nonetheless, the distribution of cisco in the gravid season provides further evidence that cisco could spawn throughout Lake Michigan as suggested in Goodyear et al. [60].

## Management implications

Globally, setting realistic goals for restoration and conservation in the face anthropogenic-driven environmental changes can prove challenging [2]. Similar to many aquatic ecosystems around the world, watershed land covers and species compositions across the Great Lakes have dramatically changed from their historical states [70]. Since 2010, the U.S. Environmental Protection Agency has invested more than $2.4 billion dollars towards improving water quality, restoring habitat and native species, and reducing the impact of exotic species (see: https://www.glri.us/action-plan). Cisco restoration in Lake Michigan remains in the early planning stages, as managers consider what strategies would be most effective and seek improved understanding of how cisco restoration may increase the biological integrity of the fish community, an overarching goal of Lake Michigan fishery managers [71]. Our work informs managers and

future possible restoration strategies by describing the historical habitat use of cisco and suggesting where a fully restored Lake Michigan population should be abundant. We highlight the importance of embayment habitats as the historical fishery-independent and fishery data revealed that the highest densities of cisco occurred in Green Bay and Grand Traverse Bay, from spring to fall.

The seasonal migration pattern identified in this study suggests that a restored cisco population can diversify the food web by occupying different habitats from those of alewife and rainbow smelt, the dominant exotic prey fishes in pelagic waters of Lake Michigan. The analyses suggest cisco migrated to deeper waters during summer, likely in search of cooler, and more optimal water temperatures. In contrast, Wells [72] reported that the alewife was concentrated in waters with bottom depths of 55–91 m over winter and migrated to nearshore waters (bottom depth <27 m) between spring and fall while the rainbow smelt stayed in nearshore waters throughout the year. However, no evaluation can be made on the extent to which cisco migrated from embayments to the main basin in summer, when optimal water temperatures would have been more plentiful in the main basin.

This paper provides a benchmark for historical habitat use of adult ciscoes, but further research is needed to understand the factors that may be impeding an even more rapid recovery of cisco in Lake Michigan. Over the past decade, cisco numbers have been increasing, with the highest density occurring in and around Grand Traverse Bay, but very low densities persist in Green Bay [59]. One ongoing research question is how ciscoes today compare to those (i.e., *C. artedi artedi*) in this study. For example, some morphological differences are apparent [23, 73]. Functionally, the large ciscoes in Grand Traverse Bay today feed primarily on prey fish [74], whereas historically ciscoes fed primarily on plankton [26, 75]. Managers are evaluating whether extant ciscoes in Lake Michigan will continue to expand and diversify or if other management actions are needed to increase both the abundance and diversification of ciscoes to enhance their recovery and long-term stability [16]. Broadly speaking, habitat uses of cisco for spawning and at larval and juvenile stages remain poorly understood in Lake Michigan and in the other Great Lakes, although Oyadomari and Auer [76] described the habitat use of larval ciscoes in Lake Superior. A more complete description and understanding for habitat use of cisco at each life stage would reveal whether the Lake Michigan ecosystem can still provide habitat connectivity through ontogeny and sustain abundant cisco populations in key historical regions, such as Green Bay.

## Supporting information

**S1 Datasets. The *Fulmar* and commercial catch datasets used in this study.**
(XLSX)

## Acknowledgments

We thank Tara Bell, Sofia Dabrowski, and Scott Nelson for retrieving archived documents, Darren Kirkendall and Wendylee Stott for retrieving and entering data recorded on scale envelopes, and Gust Annis, Matthew Herbert, and Hanna Schaefer for discussion about historical cisco spawning habitat in Lake Michigan. With their multi-decadal experience in this field, Charles Madenjian and Timothy DeSorcie helped us understand the operation of *Fulmar* surveys. Matthew Herbert also provided comments for an earlier version of this manuscript. Any use of trade, product, or firm names is for descriptive purposes only and does not imply endorsement by the U.S. Government.

## Author Contributions

**Conceptualization:** David B. Bunnell, Randy L. Eshenroder.

**Data curation:** Yu-Chun Kao, David B. Bunnell, Randy L. Eshenroder, Devin N. Murray.

**Formal analysis:** Yu-Chun Kao, David B. Bunnell.

**Funding acquisition:** Yu-Chun Kao, David B. Bunnell.

**Methodology:** Yu-Chun Kao, David B. Bunnell.

**Project administration:** David B. Bunnell.

**Resources:** David B. Bunnell.

**Supervision:** Yu-Chun Kao.

**Visualization:** Yu-Chun Kao.

**Writing – original draft:** Yu-Chun Kao, Devin N. Murray.

**Writing – review & editing:** Yu-Chun Kao, David B. Bunnell, Randy L. Eshenroder, Devin N. Murray.

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
