## [Decision Letter · Decision Letter 0]

3 Feb 2020

PONE-D-19-34710

Describing historical habitat use of a native fish—cisco (Coregonus artedi)—in Lake Michigan between 1930 and 1932

PLOS ONE

Dear Dr. Kao,

Thank you for submitting your manuscript to PLOS ONE. After careful consideration, we feel that it has merit but does not fully meet PLOS ONE’s publication criteria as it currently stands. Therefore, we invite you to submit a revised version of the manuscript that addresses the points raised during the review process.

ACADEMIC EDITOR: I read with interest your MS and found it very well written. However, you need to address the queries by reviewer #1 which I found to be relevant and will improve your MS. I look forward to reading the corrected version.

We would appreciate receiving your revised manuscript by Mar 19 2020 11:59PM. To enhance the reproducibility of your results, we recommend that if applicable you deposit your laboratory protocols in protocols.io, where a protocol can be assigned its own identifier (DOI) such that it can be cited independently in the future. For instructions see: http://journals.plos.org/plosone/s/submission-guidelines#loc-laboratory-protocols

We look forward to receiving your revised manuscript.

Kind regards,

Ismael Aaron Kimirei, Ph.D.

Academic Editor

PLOS ONE

Journal Requirements:

2. We note that Figure [2] includes an image of a [patient / participant / in the study]. 

3. We note that [Figure(s) #] in your submission contain [map/satellite] images which may be copyrighted. All PLOS content is published under the Creative Commons Attribution License (CC BY 4.0), which means that the manuscript, images, and Supporting Information files will be freely available online, and any third party is permitted to access, download, copy, distribute, and use these materials in any way, even commercially, with proper attribution. For these reasons, we cannot publish previously copyrighted maps or satellite images created using proprietary data, such as Google software (Google Maps, Street View, and Earth). For more information, see our copyright guidelines: http://journals.plos.org/plosone/s/licenses-and-copyright.

1.    You may seek permission from the original copyright holder of Figure(s) [#] to publish the content specifically under the CC BY 4.0 license. 

Reviewers' comments:

Reviewer's Responses to Questions

**Comments to the Author**

1. Is the manuscript technically sound, and do the data support the conclusions?

Reviewer #1: Yes

Reviewer #2: Yes

2. Has the statistical analysis been performed appropriately and rigorously? 

Reviewer #1: Yes

Reviewer #2: Yes

3. Have the authors made all data underlying the findings in their manuscript fully available?

Reviewer #1: Yes

Reviewer #2: Yes

4. Is the manuscript presented in an intelligible fashion and written in standard English?

Reviewer #1: Yes

Reviewer #2: Yes

5. Review Comments to the Author

Reviewer #1: Review of Yu-Chun Kao et al. Describing historical habitat use of a native fish – Cisco – In Lake Michigan.

General Comment: The manuscript is very well written, utilizes a very important data set, and addresses an important issue (cisco in Lake Michigan). Besides the management implications section, the manuscript can be accepted as submitted. However, the specific comments are mainly relative to the management implications section and must be addressed prior to acceptance. This review and the specific comments below are directly from a Michigan manager and the authors statements, albeit brief, have implications on framing the importance of the science in the management for cisco.

I have provided specific wording changes and suggest that the manuscript not be accepted unless those changes are incorporated. In conclusion, the introduction, methods, results, and most of the manuscript is of extremely high quality and will be a very important contribution to the Great Lakes science and management. However, none of the authors are fisheries managers and therefore they should recast the implications of this work from the issue of the ‘form’ of cisco as being the next important step to other, more pressing questions, that are critically important to fisheries managers. Simply put, the authors provide strong evidence that the bays, both Grand Traverse and Green Bay, are extremely important to both the recovery and long-term stability of cisco in Lake Michigan. Managers are extremely interested in why the existing stocks in Lake Michigan are experiencing a rapid expansion in Grand Traverse Bay and not Green Bay. The authors work in this manuscript can support the importance of that question and drive applied research as to the potential limiting factors (differences in food, spawning or nursery habitat, productivity, food web implications, etc) and should be restated as such.

Specific comments:

Line 399: Modify to specify that the spring movement is likely for feeding. Currently reads as though both the spring and fall movement inshore is for spawning.

Lines 521-522. “One ongoing question is how the cisco that are abundant today compare to those (i.e., C. artedi artedi) in this study.” This is an important question for researchers, but less important to fisheries managers. Only two decades ago, cisco were relatively undetectable in Lake Michigan. With the resurgence only recently occurring, managers have much more pressing questions about factors contributing to the resurgence, potential impediments to the recovery, interactions with other Coregonines (e.g., lake whitefish), interactions with other predators (e.g., salmonines important to managers), availability of critical habitats for early life stages, factors limiting the spatial expansion of the existing stocks (e.g., from Grand Traverse to Green Bay), development of population models as pressure to capitalize on cisco from the commercial fishery increases with perceived increases in cisco abundances, etc. A major disconnect in Lake Michigan is the emphasis of the research community on the forms of cisco, both historically and recently, while ignoring the questions important to the management community. The authors should either classify this statement as a question important to researchers and not necessarily management, or better frame this question and its importance to the other suite of management questions that will be important to supporting the recovery of cisco in Lake Michigan.

Lines 522-523. For example, some morphological differences are apparent [23]. Need to include Yule et al. 2013. Yule, D.L., S.A. Moore, M.P. Ebener, R.M. Claramunt, T.C. Pratt, L.L. Salawater, and M.J. Connerton. 2013. Morphometric variation among spawning cisco aggregations in the Laurentian Great Lakes: are historic forms still present? Advances in Limnology 64: 119-132.

Lines 523-527. Reword these sentences as follows: Functionally, the large ciscoes in Grand Traverse Bay today can utilize both invertebrates and prey fish [73], whereas historically ciscoes fed primarily on plankton [26, 74]. Managers are evaluating whether the cisco stocks that currently exist will continue to expand and diversify or if other management actions are needed to increase both the abundance and diversification of cisco stocks to enhance their recovery and long-term stability [16]. Delete the following", and different forms might be expected to occupy different habitats, at least seasonally or at different life stages."

Reviewer #2: This manuscript is exceptional, and I believe it should be published as is. There are minor grammatical errors that need to be cleaned up before publication. This analysis of an important and historic fishery and habitat use will have a positive impact on restoration of coregonids in Lake Michigan.

6. PLOS authors have the option to publish the peer review history of their article (what does this mean?). If published, this will include your full peer review and any attached files.

Reviewer #1: No

Reviewer #2: No

---

## [Author Response · Author response to Decision Letter 0]

18 Mar 2020

General Comment: The manuscript is very well written, utilizes a very important data set, and addresses an important issue (cisco in Lake Michigan). Besides the management implications section, the manuscript can be accepted as submitted. However, the specific comments are mainly relative to the management implications section and must be addressed prior to acceptance. This review and the specific comments below are directly from a Michigan manager and the authors statements, albeit brief, have implications on framing the importance of the science in the management for cisco.

I have provided specific wording changes and suggest that the manuscript not be accepted unless those changes are incorporated. In conclusion, the introduction, methods, results, and most of the manuscript is of extremely high quality and will be a very important contribution to the Great Lakes science and management. However, none of the authors are fisheries managers and therefore they should recast the implications of this work from the issue of the ‘form’ of cisco as being the next important step to other, more pressing questions, that are critically important to fisheries managers. Simply put, the authors provide strong evidence that the bays, both Grand Traverse and Green Bay, are extremely important to both the recovery and long-term stability of cisco in Lake Michigan. Managers are extremely interested in why the existing stocks in Lake Michigan are experiencing a rapid expansion in Grand Traverse Bay and not Green Bay. The authors work in this manuscript can support the importance of that question and drive applied research as to the potential limiting factors (differences in food, spawning or nursery habitat, productivity, food web implications, etc) and should be restated as such.

Specific comments:

Line 399: Modify to specify that the spring movement is likely for feeding. Currently reads as though both the spring and fall movement inshore is for spawning.

Response: After reading this paragraph again, we revised the sentence by deleting “when spawning occurs” (line 430 in the revised manuscript) and revised a sentence conveying similar information in the Abstract (line 28 in the revised manuscript).

Lines 521-522. “One ongoing question is how the cisco that are abundant today compare to those (i.e., C. artedi artedi) in this study.” This is an important question for researchers, but less important to fisheries managers. Only two decades ago, cisco were relatively undetectable in Lake Michigan. With the resurgence only recently occurring, managers have much more pressing questions about factors contributing to the resurgence, potential impediments to the recovery, interactions with other Coregonines (e.g., lake whitefish), interactions with other predators (e.g., salmonines important to managers), availability of critical habitats for early life stages, factors limiting the spatial expansion of the existing stocks (e.g., from Grand Traverse to Green Bay), development of population models as pressure to capitalize on cisco from the commercial fishery increases with perceived increases in cisco abundances, etc. A major disconnect in Lake Michigan is the emphasis of the research community on the forms of cisco, both historically and recently, while ignoring the questions important to the management community. The authors should either classify this statement as a question important to researchers and not necessarily management, or better frame this question and its importance to the other suite of management questions that will be important to supporting the recovery of cisco in Lake Michigan.

Response: Thanks for this comment. We made it clear this is a question interesting to the research community (line 553 in the revised manuscript).

Lines 522-523. For example, some morphological differences are apparent [23]. Need to include Yule et al. 2013. Yule, D.L., S.A. Moore, M.P. Ebener, R.M. Claramunt, T.C. Pratt, L.L. Salawater, and M.J. Connerton. 2013. Morphometric variation among spawning cisco aggregations in the Laurentian Great Lakes: are historic forms still present? Advances in Limnology 64: 119-132.

Response: We added this reference (reference #73).

Lines 523-527. Reword these sentences as follows: Functionally, the large ciscoes in Grand Traverse Bay today can utilize both invertebrates and prey fish [73], whereas historically ciscoes fed primarily on plankton [26, 74]. Managers are evaluating whether the cisco stocks that currently exist will continue to expand and diversify or if other management actions are needed to increase both the abundance and diversification of cisco stocks to enhance their recovery and long-term stability [16]. Delete the following", and different forms might be expected to occupy different habitats, at least seasonally or at different life stages."

Response: We revised the sentences (lines 557–562 in the revised manuscript).

Reviewer #2: This manuscript is exceptional, and I believe it should be published as is. There are minor grammatical errors that need to be cleaned up before publication. This analysis of an important and historic fishery and habitat use will have a positive impact on restoration of coregonids in Lake Michigan.

Response: Thank you!

---

## [Editor Report · Decision Letter 1]

24 Mar 2020

Describing historical habitat use of a native fish—cisco (Coregonus artedi)—in Lake Michigan between 1930 and 1932

PONE-D-19-34710R1

Dear Dr. Kao,

We are pleased to inform you that your manuscript has been judged scientifically suitable for publication and will be formally accepted for publication once it complies with all outstanding technical requirements.

With kind regards,

Ismael Aaron Kimirei, Ph.D.

Academic Editor

PLOS ONE

Additional Editor Comments (optional):

In LN 494, please replace "....it was assumed that cisco more likely inhabited waters....." with "....it was assumed that, more likely, Cisco inhabited waters...." 
---

## [Editor Report · Acceptance letter]

26 Mar 2020

PONE-D-19-34710R1 

Describing historical habitat use of a native fish—cisco (Coregonus artedi)—in Lake Michigan between 1930 and 1932 

Dear Dr. Kao:

I am pleased to inform you that your manuscript has been deemed suitable for publication in PLOS ONE. Congratulations! Your manuscript is now with our production department. 

With kind regards,

on behalf of

Dr. Ismael Aaron Kimirei 

Academic Editor

PLOS ONE